# Autonomous Vehicles Mapping Plitvice Lakes National Park, Croatia

**Nadir Kapetanović** [1,*] ![ID], **Branko Kordić** [2] ![ID], **Antonio Vasilijević** [1], **Đula Nađ** [1] ![ID] and **Nikola Mišković** [1]

[1] Laboratory for Underwater Systems and Technologies (LABUST), Faculty of Electrical Engineering and Computing, University of Zagreb, Unska 3, 10000 Zagreb, Croatia; antonio.vasilijevic@fer.hr (A.V.); dula.nad@fer.hr (Đ.N.); nikola.miskovic@fer.hr (N.M.)

[2] Croatian Geological Survey, Sachsova 2, 10000 Zagreb, Croatia; bkordic@hgi-cgs.hr

[*] Correspondence: nadir.kapetanovic@fer.hr

**Abstract:** Plitvice Lakes National Park is the largest national park in Croatia and also the oldest from 1949. It was added to the UNESCO World Natural Heritage List in 1979, due to the unique physicochemical and biological conditions that have led to the creation of 16 named and several smaller unnamed lakes, which are cascading one into the next. Previous scientific research proved that the increased amount of dissolved organic matter (pollution) stops the travertine processes on Plitvice Lakes. Therefore, this complex, dynamic but also fragile geological, biological and hydrological system required a comprehensive limnological survey. Thirteen of the sixteen lakes mentioned above were initially surveyed from the air by an unmanned aircraft equipped with a survey grade GNSS and a full frame high-resolution full-screen camera. From these recordings, a georeferenced, high-resolution orthophoto was generated, on which the following surveys by a multibeam sonar depended. It is important to mention that this was the first time that these lakes had ever been surveyed both with the multibeam sonar technique and with such a high-resolution camera. Due to the fact that these thirteen lakes are difficult to reach and often too shallow for a boat-mounted sonar, a special autonomous surface vehicle was developed. The lakes were surveyed by the autonomous surface vehicle mounted with a multibeam sonar to create detailed bathymetric models of the lakes. The missions were planned for the surface vehicle based on the orthophoto from the preliminary studies. A detailed description of the methodology used to survey the different lakes is given here. In addition, the resulting high-resolution bathymetric maps are presented and analysed together with an overview of average, maximum depths and number of data points. Numerous interesting depressions, which are phenomena consistent with previous studies of Plitvice Lakes, are noted at the lake beds and their causes are discussed. This study shows the huge potential of remote sensing technologies integrated into autonomous vehicles in terms of much faster surveys, several orders of magnitude more data points (compared to manual surveys of a few decades ago), as well as data accuracy, precision and georeferencing.

**Keywords:** autonomous surface vehicle; unmanned aerial vehicle; multibeam sonar; bathymetry; photogrammetry

## 1. Introduction

Plitvice Lakes National Park (cro. Nacionalni Park Plitvička jezera) is the oldest and largest national park in the Republic of Croatia. The park is located in the mountainous region of Croatia, as the map in Figure 1 shows. With its exceptional natural beauty, this area has always attracted nature lovers, and as early as 8 April 1949 it was declared the first national park in Croatia. Dominik Vukasović,

the parson of the Plitvice Lakes area in 1777, was the first to mention the name "Plitvička jezera". The name is most likely derived from the Croatian word for shallow water (pličina, or plitvak) [1,2].

The process of tufa formation, which results in the building of the tufa, or travertine barriers and thus to the creation of the lakes, is the outstanding universal value for which Plitvice Lakes were internationally recognised on 26 October 1979 when they wew inscribed on the UNESCO World Heritage List UNESCO. In 1997, the boundaries of the National Park were extended, and today it covers an area of almost 300 km$^2$ [3].

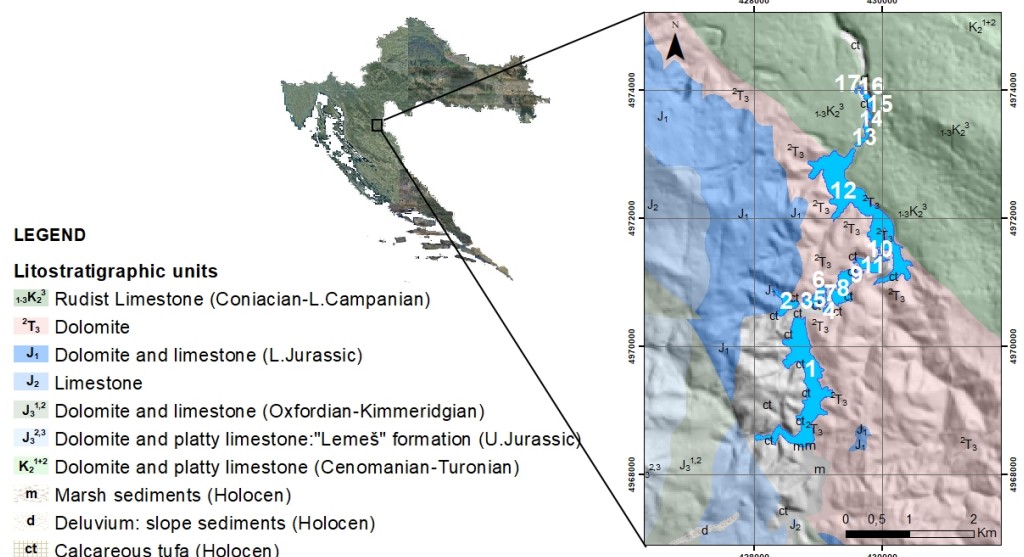

**Figure 1.** A simplified geological map of the Plitvice National Park modified after [4]. The shape of the lakes and the surrounding terrain closely reflect the underlying geological structure. The bedrock of the Upper Lakes (denoted 1–12) is mainly the Upper Triassic dolomite that is relatively impermeable and retain water, making Upper lakes resting in gently sloping valley. The bedrock of the Lower Lakes (dentoed 13–16) is the Upper Cretaceous limestone that was susceptible to karstification processes, making Lower lakes narrower and situated in deep canyon. Courtesy of Josip Barbača from Croatian Geological Survey.

The lake system consists of 17 named and several smaller unnamed lakes, which cascade into each other. Due to the geological substrate and the characteristic hydrogeological conditions, the lake system was divided into the Upper and Lower lakes. The twelve lakes that form Upper Lakes are Prošćansko jezero (marked 1 in Figure 1), Ciginovac (marked 2 in Figure 1), Okrugljak (marked 3 in Figure 1), Batinovac (marked 4 in Figure 1). Veliko jezero (marked 5 in Figure 1), Malo jezero (in Figure 1 marked with 6), Vir (in Figure 1 marked with 7). Galovac (marked 8 in Figure 1, represented in Figure 2), Milino jezero (identified by 9 in Figure 1), Gradinsko jezero (identified by 10 in Figure 1 is 10), Burgeti (in Figure 1 is 11) and Kozjak (in Figure 1 is 12). These lakes were formed on impermeable dolomite rock and are larger, with more articulated and gentler shores than the lower lakes. The Lower lakes, consisting of the lakes Milanovac (marked 13 in Figure 1), Gavanovac (marked 14 in Figure 1), Kaluđerovac (marked 15 in Figure 1), Kaluđerovac (marked 16 in Figure 1) and Novakovića Brod (shown as 17 in Figure 1), were formed in permeable limestone substrate cut into a deep canyon with steep cliffs. The lakes end in the impressive Sastavci waterfalls, with the Korana River springing from the base of the falls.

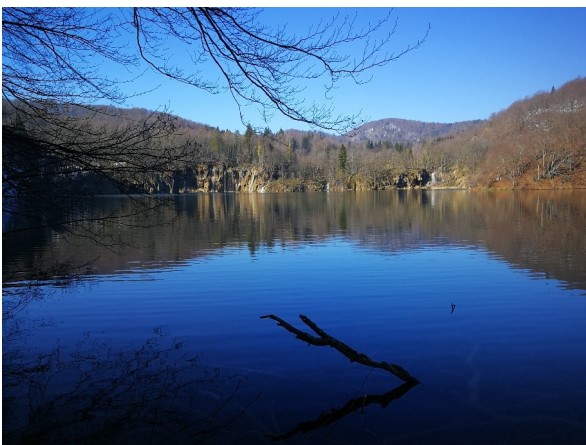

**Figure 2.** View of the lake Galovac. Courtesy of Nadir Kapetanović.

In Plitvice Lakes National Park there are partitions between the lakes, over which waterfalls and cascades fall, that occur in a special natural way. Under certain physicochemical and biological conditions, they form tufa and settle on the lake bottom and on submerged objects. Tufa forms underwater thresholds and barriers that rise above the water and grow steadily in height and breadth. This constant tufa formation with warm climate, lush vegetation and undisturbed natural balance was the first factor that made Plitvice Lakes part of the World Natural Heritage [5].

Tufa is deposited on the bottom of Plitvice Lakes in the form of microscopic crystals that cover the lake bottom with a thick layer, thus preventing water loss through the perforated karst carrier. Thanks to this, the large Upper Lakes (Lake Prošćansko and Lake Kozjak), which lie on dolomite substrate, do not lose water, but after the beginning of the flow of Korana River, water is lost underground because the travertine processes stop [5].

Today, in the area of Plitvice Lakes National Park it is possible to find places where travertine is absent or reduced in intensity. Scientific studies have shown that the increased amount of dissolved organic matter (pollution) stops the travertine creation processes at Plitvice Lakes [1]. Thus, the inevitable process of eutrophication or the aging process of a lake takes place. This is a natural process that usually lasts thousands of years. However, it can be significantly accelerated by human activities (agriculture, livestock, tourism, sewage, settlements and hotels) [5].

Eutrophication is the process of enriching water with organic nutrients that promote the growth of aquatic plants such as planktonic algae, bottom algae and other aquatic plants. Plitvice Lakes are covered with swamp vegetation and lake bottoms are covered with underwater meadows. All this vegetation lays on travertine barriers. Sometimes there are even trees growing on the edges of waterfalls whose weight endangers waterfalls' stability and threatens to cause waterfalls to collapse. The eutrophication itself is a normal natural aging process of the lakes over thousands of years, while anthropogenic eutrophication is caused by human activities and can destroy the aquatic ecosystem in a very short time. Obviously, Plitvice lakes have been affected by the process of anthropogenic eutrophication in recent decades [5].

Depth measurements and the characterisation/classification of the lake bottom are of utmost importance for research of tufa formation and eutrophication processes in the lakes. Major Franz (Franjo) Bach was the first to carry out depth measurements of Plitvice Lakes in 1850 [1]. His work was improved by contributions from Franić in 1910 [2], Gavazzi in 1919 [6], and Petrik in 1958 [7]. All these studies involved measuring depths directly by a rock tied to a wire, which was then thrown from a small boat at as many points as possible. These measurements were very valuable for their time, but they were not georeferenced and the depth measurements were much less accurate compared to modern indirect depth measurement methods.

It is only in the last 20 to 30 years that we have been able to explore and map the underwater world of the Earth, mainly through technical advances such as acoustic remote sensing. Models based

on acoustic data can be used to estimate how underwater locations have changed both recently and far in the past but also to predict how they might change in the future [8,9]. These models can then be used as powerful tools in public commitment to environmental protection and conservation. Nothing has advanced in underwater technologies and research areas as much as localisation and environmental imaging devices. Recently, photogrammetry, photo modelling, simultaneous localisation and navigation (SLAM), organised light processing, multibeam and numerous other acoustic sensing techniques have become ubiquitous [10–13]. In 2007 and 2010 Pribičević and his colleagues bring modern acoustic methods of bathymetric measurements of the two largest lakes in Plitvice, namely Kozjak and Prošćansko lake [5,14].

Robotics is developing into a successful tool in the exploration of shallow waters and offers a wide range of surveying possibilities (2.5D positioning or landscape mapping without excavation) [15]. Human operators/data collectors combine high versatility, intelligent coordination and a wide variety of manual skills to reduce operating costs. However, these human advantages tend to disappear as the area to be explored grows or the time required for field operations becomes shorter. In addition, robots can endure longer survey missions without the fatigue that humans are prone to, they can move faster, deeper and higher, they do not risk human injury or fatalities and their deployment is more cost-effective in the long run. They can also be used very well in the cold as well as low visibility/turbid environments and they usually have an array of localisation/survey sensors whose sensory range is much greater than that of a human and/or DSLR camera.

A few decades ago, precise robot localisation was a major challenge and a great stepping stone to the use of autonomous robots for detailed and precise surveying missions in unstructured environments. Navigation and localisation are among the most demanding problems in the production of autonomous vehicles. However, the use of surface vessels based on a combination of Global Navigation Satellite System (GNSS), Inertial Measurement Unit (IMU) and/or Dopller Velocity Logger (DVL) navigation in shallow underwater archaeology can solve these problems. In contrast to the slow acoustic communication channel required underwater, a surface vessel often provides a fast wireless communication link to the base.

Robot localisation equipment and algorithms have improved significantly, allowing the experts using the robots to acquire acoustic/visual data with precise position markers and feed it directly into commercially available bathymetry and/or photogrammetry software. All of the technological improvements mentioned above have led to a significant increase in the use of robotic systems for surveying underwater sites, either with or instead of divers. These robotic systems include remotely operated vehicles (ROVs), autonomous surface vehicles (ASVs), autonomous underwater vehicles (AUVs) and unmanned aerial vehicles (UAVs) for inspections in shallow water. The applications of these vehicles range from underwater archeology, biology, ecology, geology, hydrography, safety and many others.

There are many understandable logistical limitations to working in Plitvice Lakes National Park. Due to the minimisation of the carbon footprint that people leave behind in the National Park, any type of gasoline-powered vehicles are strictly prohibited, so only electrically powered vehicles come into play. Even access to the lakes with all the research equipment can sometimes be extremely strenuous. The morphology of the terrain is very rough and steep. Around some lakes there are forest paths only a few meters wide, while some of the lakes are crossed by quite narrow wooden footbridges. The installation of a multibeam sonar on a small boat with electric motor drive seems to be a simple but effective solution. However, the main problem with this human-operated surveying approach is that the planning and execution of surveying missions is prone to human error and could therefore result in some parts of the lake bottom not to be covered by the sonar.

All this leads to the conclusion that it seemed ideal for the research purposes of such a national park to design and use a small, lightweight, electrically driven ASV with engines for such purposes. It would be able to be used by only two people in the lakes, could be easily moved around the lakes on a small trolley, could be transported in a passenger car and could be manufactured with a low-cost,

low-power electrically driven thrusters since the water of the lakes would not interfere with the movement of the vehicle. Multibeam sonar mounted on such a vehicle could provide high-resolution acoustic data that could be used for hydrographic, geological and biological research purposes. The coverage of the lake bottom by the sonar would be ensured by the mission plan, which the vehicle would carry out completely autonomously.

This article encompasses the methodology and results of the authors' recent activities concerning the use of autonomous vehicles for acoustic recording (bathymetric mapping) of most of the Plitvice lakes. The main contribution of this article is the fact that for the first time in history these lakes were recorded by an autonomous vehicle using a multibeam sonar. The authors used an ASV specifically designed for bathymetric surveys of lakes and shallow marine waters and a UAV for visual data acquisition. The results of the postprocessed collected data are presented as a bathymetric map/2.5D model and/or visual recordings by the ASV and the UAV. During the monitoring missions in 2019, a total of 11 of 16 lakes were surveyed with a resolution of 20 cm, together with the lakes Kozjak and Prošćansko, which were previously surveyed by one of the authors in [5,16]. The methodology of covering regular and irregularly shaped lakes by sonar swaths is presented together with digital elevation maps (DEMs) for the 11 surveyed lakes. In addition, numerous depressions in tufa at the lake bottom were detected in bathymetric but also in backscatter data. It is assumed that these are natural karst terrain formations. Their detailed exploration in the future with various other sensors such as sub-bottom profilers, cameras mounted on an AUV or an ROV as well as acoustic Doppler current profilers (ADCPs) is necessary for a deeper understanding of this intricate and geologically very dynamic lake system.

The rest of this article is structured as follows: a general overview of the authors' activities in recent years, including research and technical projects, and the publications contained therein can be found in Section 2. The autonomous robotic vehicles used for underwater photography and their sensors are presented in Section 3. The methodology of remote sensing work applied to geological applications is presented in Section 4. Section 5 presents the results of the Plitvice Lakes' bathymetric survey missions. The results and future work are discussed in Section 6. Concluding remarks are given in Section 7.

## 2. Prior Work

### 2.1. Limnological Research of Plitvice Lakes

Gerhard Mercator was the first cartographer to map Plitvice Lakes in the 16th century as a single lake without a name. In the following century, a map was published in 1664 that depicted four nameless lakes southwest of the upper reaches of the Korana River [1,2]. Major Franz (Franjo) Bach is considered to be the first who carried out a limnological study of the Plitvice Lakes in 1850 [1]. Following Bach, Dragutin Franić calculated the surfaces of the lakes with the help of cadastral and geographic maps at the scale 1:25,000 [1,2,7].

In 1919 Artur Gavazzi conducted the first comprehensive limnological study of Plitvice Lakes in [1,6]. In his study, Gavazzi included the relative altitudes of the lakes, their absolute altitudes under low water levels and the calculated areas and volumes of the lakes. Furthermore, he made bathymetric maps of the lakes Prošćansko and Kozjak.

After Plitvice Lakes were proclaimed a national park (NP) in 1949, the first survey study from 1951 to 1954 was financed by NP and published in [7]. Petrik M. led the studies, in the course of which almost 5400 measurements of the sea depths were collected. On the basis of these measurements he determined isobaths of lake depths. He also showed in detail the submerged barriers found in the lakes Malo jezero and Kozjak. Moreover, his study includes absolute lake altitudes, maximum lake depths and estimated areas and volumes [1].

Petrik also noted that his lake depth data differed from the data in the studies of Franić [2] and Gavazzi [6]. He concluded that this is due to the rapid geological changes that occur on the lake

floor within a few decades. Furthermore, he noted that due to these changes, some lakes even merge into a single lake and that the depth of the lakes increases when moving downstream in the Plitvice Lakes system.

Until the 21st century and studies found in [5,14,16] the limnological studies of Plitvice Lakes were performed with the methods of direct depth measurement. New technologies applied in hydrography and geodesy, namely the development of high-precision GNSS units, multibeam sonars, ubiquitous unmanned aerial vehicles and photogrammetric 3D visual reconstruction software, enabled much faster limnological studies with a huge amount of high-resolution data georeferenced with an accuracy of a few centimeters.

The bathymetry of the two largest lakes, lake Prošćansko and lake Kozjak, is presented in [5,14,16]. Although the authors investigated the two largest lakes, their published results corresponded mainly to the lake Prošćansko. The authors used an advanced ultrasonic echo sounder measurement technique in combination with a Global Position System (GNSS). They used two different probes, one emitting high frequency (210 kHz) and the other low frequency (33 kHz) signals. This dual-frequency technique is based on the different behaviour of signals emitted at different frequencies. The higher frequency is reflected by the sediment surface, while the lower frequencies penetrate deeper and can thus detect different sediment structures. Raw measurements and processed bathymetric maps were delivered to the Research Institute Ivo Pevalek at Plitvice Lakes National Park. These data are used both by the researchers of the institute and by researchers from outside the institute, who carry out research at Plitvice Lakes [16]. The lakes Prošćansko and Kozjak were recorded with a multibeam interferometric sonar for further research. The survey was carried out in the classical way for acoustic methods with a boat with crew. This method could not be applied to other lakes because they are shallower and more difficult to reach.

## 2.2. Robotics Research for Remote Sensing

Laboratory for Underwater Systems and Technologies (LABUST) has participated in many research and technical projects that pushed the boundary of the state-of theart in marine robotics and remote sensing applications in numerous scientific fields. For example, in the projects TRITON [17], ADRIAS [18] and BLUEMED [19,20] they have used marine robotics as a remote sensing tool for maritime archaeology; marine ecology in the projects subCULTron [21] and e-UReady4OS [22,23]; Maritime safety in the MORUS project [24]; Maritime inspection in the HEKTOR project; Promotion of diver safety and augmenting their performance in the CADDY [25] and ADRIATIC [26] projects An overview of the geographical distribution of the sites where LABUST's marine robots were used for remote sensing is shown in Figure 3.

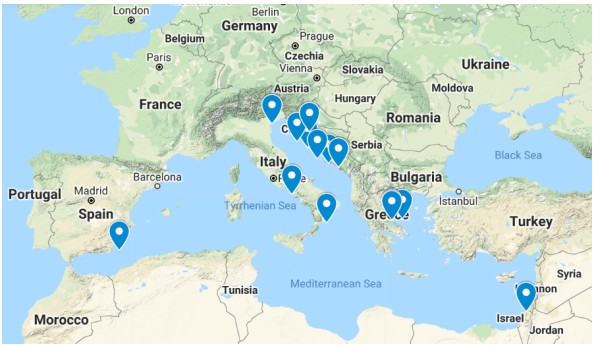

**Figure 3.** Google Map of locations at which LABUST used marine robots in remote sensing missions.

The long-term research goal of the authors is to develop the current unmanned survey/inspection missions by marine vehicles into missions that are performed by the marine vehicles in a fully autonomous manner controlled by artificial intelligence. This of course means that online sensor data processing must be developed to enable the vehicle to perceive its environment (as published

in [27–29]), as well as mission and path planning algorithms, so that the behaviour of the vehicle is responsive to the new information about its environment as published in [21,30–33]. For successful, fully autonomous reconnaissance missions it is of utmost importance that the marine vessels estimate their position accurately, as published by the authors in [22,34,35].

## 3. Equipment

A modular autonomous surface vehicle (ASV) equipped with a multibeam sonar, which was used for remote sensing survey missions, is described in more detail in Section 3.1. The unmanned aerial vehicle (UAV) that was used for the inspection is briefly described in Section 3.2.

### 3.1. Autonomous Surface Vehicle

An ASV equipped with a Norbit WBMS 400/700 KHz multibeam sonar and accompanying Applanix navigation system together with a high-precision Trimble GNSS antenna were also used to collect the acoustic data. This is one of the many application-dependent versions of the so-called Dynamic Positioning Platforms (PlaDyPos or H2Omni-X), called Dynamic Bathymetric Imaging Platform (PlaDyBath), which is shown in Figure 4. The surface vehicle was developed by the Laboratory for Underwater Systems and Technologies (LABUST), Faculty of Electrical Engineering and Computing, University of Zagreb (UNIZG-FER), Croatia, and is used for a variety of applications, from support to underwater archeology [20], as a dive monitoring platform, which enables the navigation and monitoring of divers from the surface [36], as a communication router between underwater and aerial vehicles [37], which is used in ASV swarms for long-term monitoring of the underwater environment [21], for mapping (extraction of photomosaic and bathymetry) of shallow water areas [38] and for mine countermeasures [39].

The ASV is fully actuated with four thrusters forming the X configuration. This configuration allows you to move horizontally in any orientation. The ASV has a diagonal length of 1 m, is 0.35 m high and weighs about 30 kg with the payload configuration in the experiments. The maximum speed under ideal conditions is 1 m/s. Such a configuration of the vessel is very well suited for research purposes due to its simple deployment procedure, robustness under real environmental conditions and low energy consumption [36,40]. For this special application, however, the ASV was redesigned into a catamaran form for better hydrodynamic performance, as shown in Figure 4a. The ASV PlaDyBath has so far been used for surveying and 3D modelling of underwater cultural heritage sites (UCH) in Croatia, Italy and Greece, as reported in [19,20].

Mission planning and control was carried out with the opensource software Neptus, developed by the Laboratório de Sistemas e Tecnologia Subaquática (LSTS) of the University of Porto, Portugal. It is a distributed Command and Control Infrastructure for the operation of all types of unmanned vehicles. Neptus supports the different phases of a typical mission life cycle: planning, simulation, execution and postmission analysis. Neptus can be adapted by operators to mission-specific requirements and extended by developers through a comprehensive plug-in framework. Neptus communicates with the vehicle's hardware via so-called IMC messages, which were also developed by LSTS. In the case of PlaDyBath ASV, these mission-specific IMC messages are bridged to Robot Operating System (ROS) on board the high-level mini PC, which in turn communicates with middleware microcontrollers to control the vessel's motors. The sonar is switched on both physically and via the network interface by the operator, who is connected to the vehicle via WiFi. The WBMS software used for sonar data acquisition on board the vessel was also controlled via a remote desktop over WiFi. As ASV PlaDyBath was connected to the Internet via a GSM-LTE modem, this enabled the connection to Croatian Positioning System (CROPOS), which provided a localisation accuracy of 2 cm in the horizontal plane and 4 cm in the vertical plane. This allowed the sonar data to be georeferenced with extremely high accuracy.

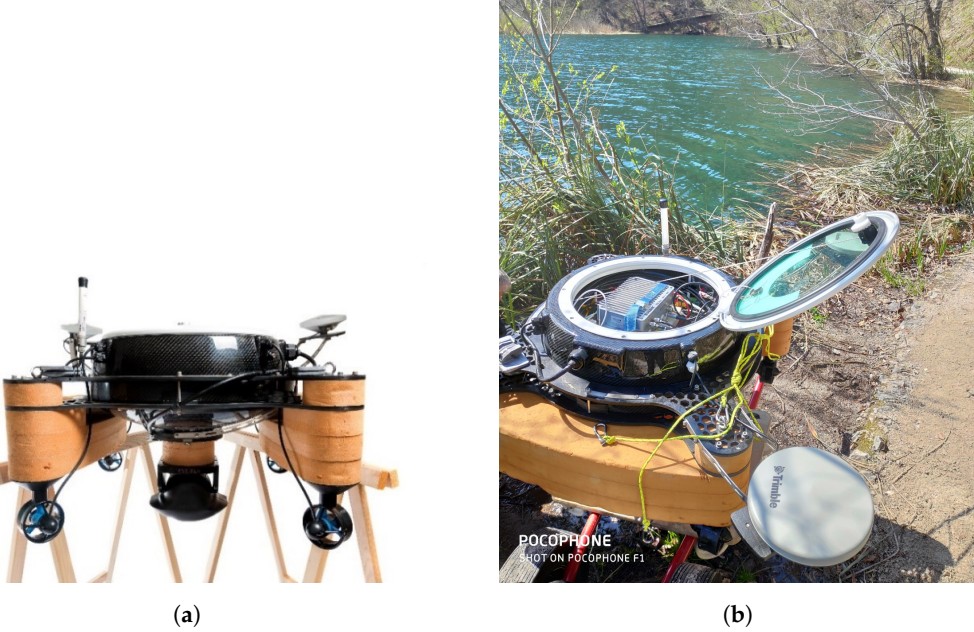

(**a**)                    (**b**)

**Figure 4.** (**a**) Autonomous surface vehicle PlaDyBath with Norbit iWBMSe multibeam sonar mounted below, Trimble GNSS antennae in the back and a WiFi antenna on the left. (**b**) ASV PlaDyBath during the monitoring mission at Gradinsko lake.

Multibeam Sonar

The Norbit iWBMSe multibeam sonar is the main sensor for ASV data acquisition, shown in Figure 4b. The sonar is integrated with the latest GNSS-assisted inertial navigation system (Applanix SurfMaster), has 80 kH bandwidth, roll stabilization, an Ethernet interface and integrated sound velocity measurement. The basic sonar features are 5–210 degrees swath, adjustable measurement sector, 10 mm resolution, 256–512 beams, 200 kHz–700 kHz nominal frequency 400 kHz, range 0.2–275 m (160 m typical at 400 kHz). Ping rate up to 60 Hz or adaptive, resolution: longitudinal × transverse standard 0.9 × 1.9 degrees at 400 kHz and 0.5 × 1.0 degrees at 700 kHz.

*3.2. Unmanned Aerial Vehicle*

A customised VTOL (Vertical TakeOff and Landing) UAV multirotor system (see Figure 5) is used as the aerial platform for all flights of our test process. The system consists of a carbon fibre tube frame with a radial array of motors powered by a high capacity Lithium Polymer (LiPo) battery [41].

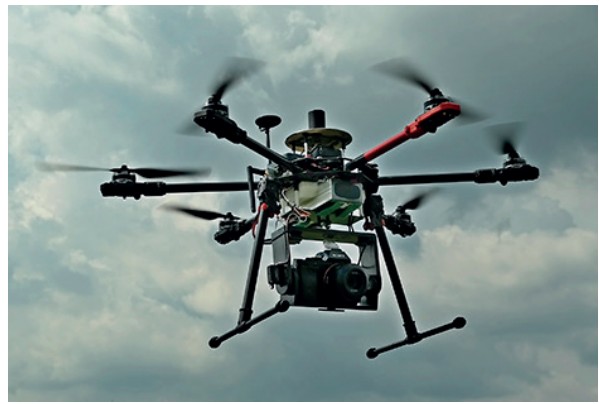

**Figure 5.** Multirotor VTOL UAV BEE-G3.

The total weight of the system is about 3.6 kg, which classifies it as a micro aerial vehicle (MAV) weighing less than 5 kg. The flight time of the system under normal conditions is about 26 min. In addition to the GNSS receiver and the camera, the platform is equipped with a Pixhawk flight controller (version: 1.8.2.), which includes additional sensors such as gyroscopes, accelerometers, 3-axis magnetometer and barometer for navigation support. The reason for using the multirotor VTOL is its ability to acquire images from different positions, its applicability in many specific tasks, the possibility of take-off and landing with the minimum required area, as well as its flight stability and operation at lower altitudes compared to light fixed-wing systems without VTOL [41].

The UAV is monitored and controlled by a ground station using dual commands via the RF link. An essential component for aerial surveying is the open source software Mission planner (Version: 1.3.50.0, Firmware: APM: Copter 3.4.4), which is used for planning purposes, control and real-time management of UAV [41].

### 3.2.1. UAV Positioning System

For this UAV the position is derived from a dual-frequency UAV GNSS receiver Septentrio AsteRx-m. Corrected positioning solutions can be obtained as PPK (Post Processing Kinematic) or RTK (Real Time Kinematic) solutions. The postprocessing of the UAV rover data with the data from the base station after the mission eliminates the need for a real-time data link between the UAV and the base station, which reduces the onboard setup and also the payload and flight time. This also eliminates a potential source of interference in the link that can occur during the RTK solution. The reduction of external effects increases the reliability of the system [41].

The system used in this research is able to determine the 3D trajectory of the platform with PPK. The raw data is stored on a memory card embedded onboard the receiver of the moving platform. The data from the base station can be stored simultaneously in raw data format or downloaded later from a network service. It is then possible to postprocess the data together with the rover data to obtain the 3D trajectory and coordinates of the events with high spatial accuracy. These coordinates represent the initial position of the captured images [41].

### 3.2.2. UAV Camera

The UAV used in this research is equipped with a Sony Alpha 7R digital camera with a 36.3 megapixel full-frame (35.9 mm × 24 mm) CMOS sensor and a Sony FE 35 mm high-quality Carl Zeiss lens.The camera was calibrated on the test field consisting of 105 evenly distributed points. The coordinates of the test field were determined by the spatial intersections taken from several occupation points measured with the TS Trimble S8. The achieved accuracy of the test field coordinates is ±0.1 mm. Phototriangulation with the self-calibration method was used to determine the parameters of the internal orientation. The phototriangulation was calculated with the BBA method (Bundle Block Adjustment) using Pix4D v4.5 software [41].

The camera is mounted on a specially designed servo-driven 2-axis gimbal with vibration dampers. In addition to position determination, the Septentrio AsteRx-m UAV-GNSS system allows time registration of the camera shutter. With this information it is possible to assign spatial coordinates to the shutter event. The method allows the direct determination of parameters of the camera position with an accuracy in the centimetre range. The receiver time stamps the shutter events of the camera in order to identify the exact time of taking the photos. These event markers are logged together with the GNSS measurements during flight on the onboard SD card for postprocessing [41].

After the flight, the data from the UAV and a base station reference receiver on the ground are postprocessed. The derived centimeter-accurate PPK position values are then embedded in the images, either directly in the EXIF data of the images or in a separate CSV file. The image coordinates contain camera trigger timing synchronization error [41].

## 4. Methodology

In March and April 2019 extensive bathymetric measurements were carried out on the lakes in Plitvice Lakes National Park in Croatia. The aim of the depth measurements of the lakes is to enable a detailed environmental monitoring of tufa formation and changes over time. The bathymetric measurements were performed by ASV PlaDyBath. In total, three of four of the Lower Lakes and eight of twelve of the Upper Lakes were surveyed.

These lakes have never been surveyed with sonar technology, with the exception of the largest lakes Kozjak and Prošćansko jezero, whose bathymetric surveys were carried out by National Park and its partners some years ago [5,14,16]. The remaining lakes were either too shallow for safe operation of the ASV PlaDyBath or the deployment of the vehicle was too complicated due to the terrain configuration. Challenging work conditions (two-person deployment of the ASV from a narrow forest trail and a makeshift workstation that had to be moved around) at Plitvice Lakes National Park are depicted in Figure 6. Processed sonar data was delivered in a form of digital elevation maps (DEMs) to National Park Plitvice for further analysis and research of the water column.

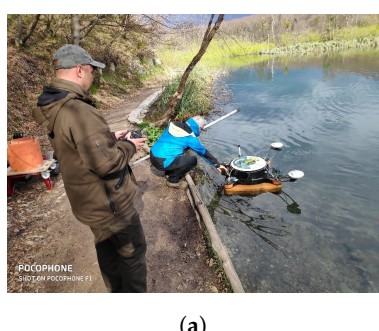 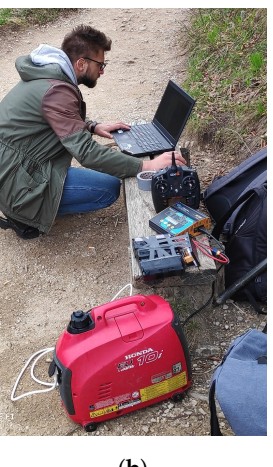

(**a**)　　　　　　　　　　　　　　　　　　　(**b**)

**Figure 6.** Challenging work conditions at Plitvice Lakes National Park. (**a**) Start of survey operations at lake Okrugljak. Two-person manual deployment of the ASV PlaDyBath from the shore of the lake. (**b**) Mission planning for the ASV PlaDyBath from a makeshift workstation.

A high-resolution georeferenced orthophoto image based on the camera images of the unmanned aerial vehicle (UAV) was used as a basis for the planning of survey missions, as shown in Figure 7. The methodology to generate such a large-scale orthophoto is partly described in [42]. This model was calculated before the bathymetry survey missions to serve as a crucial input for the safety of the ASV PlaDyBath. The outlines of the lakes were extracted from these precisely geolocalized orthophoto models with an offset of about 5–10 m from the shore to avoid shallow parts or other potentially dangerous obstacles along the shore, as shown in Figures 8a and 9a. Particularly frequent obstacles were the tree trunks that fell into the lakes and were not moved by the National Park staff, as any human intervention in the National Park is strictly prohibited. It can also be noted that the outline of the exact lake boundary does not match the map background from the map loaded in Neptus, because the resolution of the Neptus map layer is not sufficient to recognise the boundaries of the lakes.

The missions were planned as follows: Plan the first mission along the perimeter of the lake, with the sonar beams tilted 30–60° towards the shore and the swath angle at 90–120°. The rest of the lake was covered with a sonar tilt of 0° (looking directly under the vehicle) and a swath angle of 90°. The ping rate of the sonar was set to adaptive in order to obtain as much quality data as possible. This meant that the sonar pinging frequency increased in shallow areas and was automatically reduced in deeper parts of the lake. The ASV's surge speed was set at 0.7 m/s, which has proven to be a good

compromise between energy consumption and surveying time during a number of survey missions. Depending on how shallow/deep a lake is, the rest of the survey for this lake was carried out as follows.

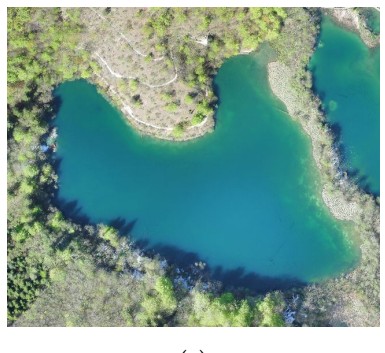 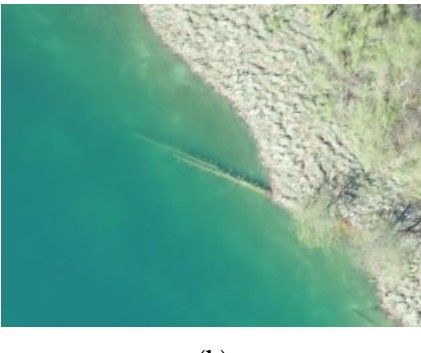

(**a**)　　　　　　　　　　　　　　　　　　　　　　　　　　　　　(**b**)

**Figure 7.** High resolution digital orthophoto image of the Okrugljak lake. (**a**) Orthophoto of the whole lake used for high precision georeferenced lake outline extraction. (**b**) Detail of a tree trunk sunk in the shallow waters of Okrugljak lake. Such areas were then excluded from the ASV's planned missions.

When a lake had a more or less convex shape, as is the case with Malo jezero shown in Figure 8a, the interior of the lake was partially covered by the so-called lawn mowing manoeuvres with variable lawn mower cross-section widths. In the shallowest parts of the lake denser transects were used, while in deeper parts of the lake wider transects were used, as shown in Figure 8b.

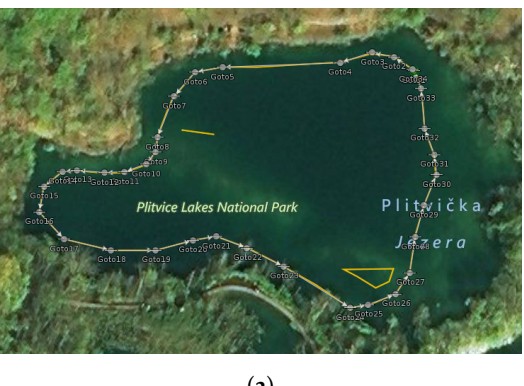 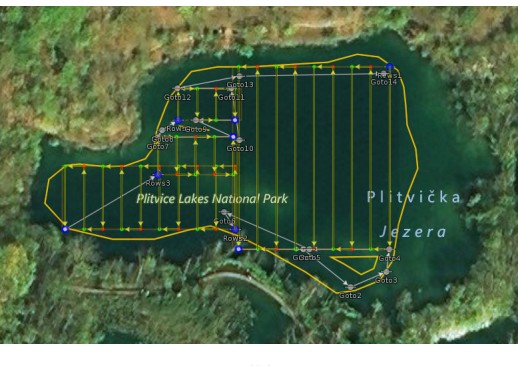

(**a**)　　　　　　　　　　　　　　　　　　　　　　　　　　　　　(**b**)

**Figure 8.** Background: Satellite imagery of Lake Malo jezero loaded in Neptus mission planning software for the ASV PlaDyBath. (**a**) The initial survey mission: The safety outline 5–10 m from the lake shore imported from a shape file based on the high resolution orthophoto with a centimeter georeferencing precision. This ensured that the ASV does not get stuck in shallow water below 0.5 m deep or that it gets stuck into many tree trunks and branches which fell into the lake. Survey mission waypoints are placed along this outline. (**b**) The rest of the lake covered by lawnmower patterns of different widths depending on the estimated depth of the lake.

If the lake was deep or had a rather nonconvex shape, as is the case with Lake Ciginovac (see Figure 9a), the interior of the lake was covered with concentric, bank-shaped missons, as shown in Figure 9b. For this type of nonautomated mission planning, it is essential that the vehicle is in WiFi range throughout the time of each circumferential pass, in order to check the depth at each waypoint it passes. The inward offset of the next concentric mission from the waypoints of the current mission is calculated as $d = 2(1 - \alpha/2)h\tan(\psi/2)$, where $d$ is the inward offset, $h$ is the depth read by the WBMS software at the specific waypoint reached by the vehicle in the current mission, $\psi$ is the sonar swath angle and $\alpha \in (0, 100\%]$ is the overlap percentage of adjacent sonar swaths. We have used $\alpha = 20\%$ and $\psi = 90°$, which gives $d = 1.8h$. We also smoothed sharp curves as much as possible to reduce the fan-out effects in the sonar footprint, which result in much sparser sonar pings on the outside and unnecessary multiple overlap of the pings on the inside of the turn towards the centre of the lake.

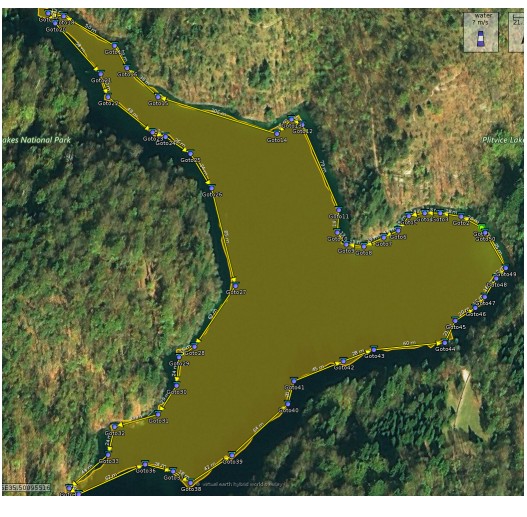
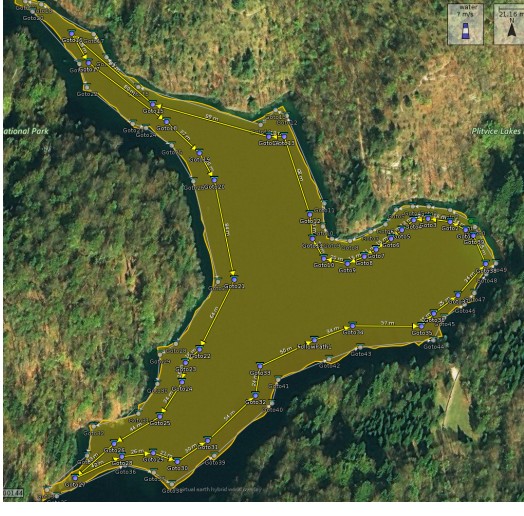

(**a**)                                                          (**b**)

**Figure 9.** Background: Satellite imagery of Lake Ciginovac loaded in Neptus mission planning software for the ASV PlaDyBath. (**a**) The initial survey mission: The safety outline 5–10 m from the lake shore imported from a shape file based on the high resolution orthophoto with a centimeter georeferencing precision. This ensured that the ASV does not get stuck in shallow water below 0.5 m deep or that it gets stuck into many tree trunks and branches which fell into the lake. Survey mission waypoints are placed along this outline. (**b**) The rest of the lake covered by concentric circumference missions. Each inner mission is offset inwards by 1.8× depth measured at the waypoints of its neighbouring outer mission.

While PlaDyBath performs its tasks autonomously, the operator processes sonar data in low resolution using the Qimera software to monitor the quality of coverage. As soon as the batteries are depleted (approx. every 1.5–2 h), the vehicle was lifted to the lakeshore, the batteries were changed and high-quality sonar data were transferred to the computer of the ASV operator. Initially, the break in surveying operations was used not only for data transfer but also for passive cooling of the Applanix SurfMaster navigation system. During the 1.5–2 h surveys on lakes, especially in shallow lakes where the adaptive pinging rate would increase significantly, it heated up very much. As mentioned before, the adaptive ping rate was used to get as much data as possible w.r.t. of the given water column depth; therefore. it was not an option to lower it. After the first visit to Plitvice Lakes National Park, a cooling system consisting of a small fan and a heat outlet was developed, which significantly improved the performance of the ASV during the second visit to Plitvice, as it solved the problem of overheating.

## 5. Results

Bathymetric data of all survey lakes were processed in the QPS Qimera software to create 2.5D manifolds representing interpolated depth profiles. The reconstruction parameters used for the reconstruction, based on sonar and IMU datasheets, were heading unceartainty: $0.3°$, roll and pitch uncertainty: $0.08°$, and the unit cell size for surface interpolation was set to 0.2 m. Bathymetric DEMs of two Plitvice Lakes are given in Figures 10 and 11. They were created in Global Mapper sotfware based on processed point cloud exported from Qimera and displayed with depth increments of 1–5 m depending on the maximum depth of each lake. Note that for each lake there is a black outline extracted from the digital orthophoto of Plitvice Lakes from the photos taken with the UAV BEE-G3. Within the outline is the generated bathymetric DEM. The white areas where the bathymetric data are missing are the shallowest parts of the lakes where the depth is less than 0.5 m and which the ASV PlaDyBath could not traverse due to safety reasons in order not to damage multibeam sonar. However, with the methodology of tilted sonar swath a good part of these shallow parts were

nevertheless recorded. Moreover, it is important to note tufa formations in the middle of the Malo lake, as Figure 11 clearly shows.

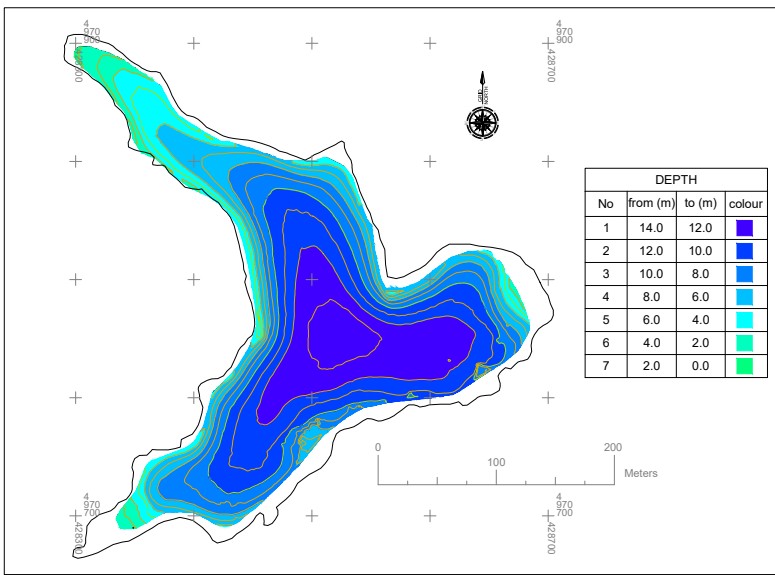

**Figure 10.** Ciginovac lake's detailed georeferenced bathymetric digital elevation map (DEM).

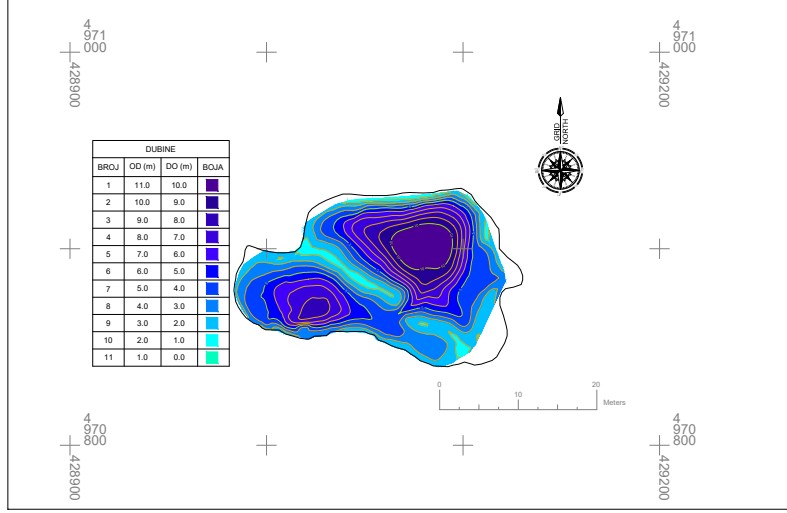

**Figure 11.** Malo jezero lake's detailed georeferenced bathymetric DEM.

In addition, the digital elevation maps (DEMs) extracted from the 2.5D bathymetry manifolds were then overlaid with the digital orthophoto of the area based on UAV camera images. Upper Lakes DEM is shown in Figure 12 and Lower Lakes' DEM is given in Figure 13. Note that the darker parts on the right side of the lakes are only the result of the standard shader in the Global Mapper software used to merge the bathymetric DEMs with photogrammetric orthophotos in Figures 12–15. It is interesting to see how the tufa-shaped lake bottom does not slope evenly from the shore to the middle of the lake. On the contrary, it contains many depressions, which are the result of the interaction between water and tufa bottom. Several detailed views of some tufa depressions in the lake bottom are shown in Figures 14 and 15. Experts in underwater acoustics and geology were consulted to explain these depth jumps. Underwater acoustics experts could not explain such measurement results with significant measured depth shifts in a small area. For this reason and because of the specific soil composition of Plitvice Lakes, geological experts were consulted. They explained that depressions at

the bottoms of the lakes are typical for karst terrains. These depressions represent sinkholes caused by the collapse of an underlying structure.

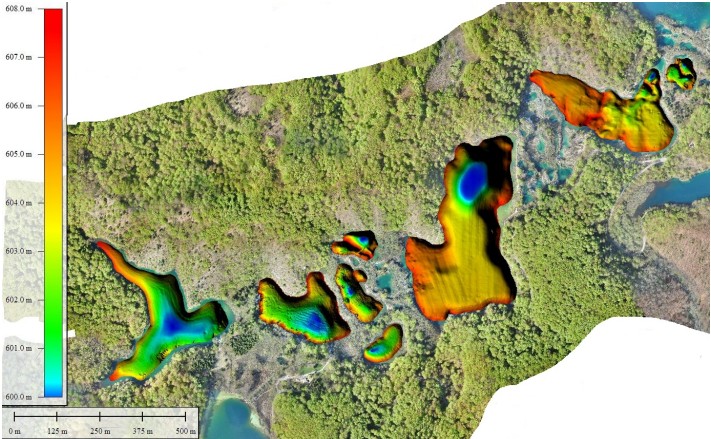

**Figure 12.** Upper Lakes' georeferenced bathymetric DEM layered over the digital orthophoto of the area. Top right to bottom left: Burget, Gradinsko, Galovac, Malo, Veliko, Burget, Ciginovac, and Okrugljak lakes.

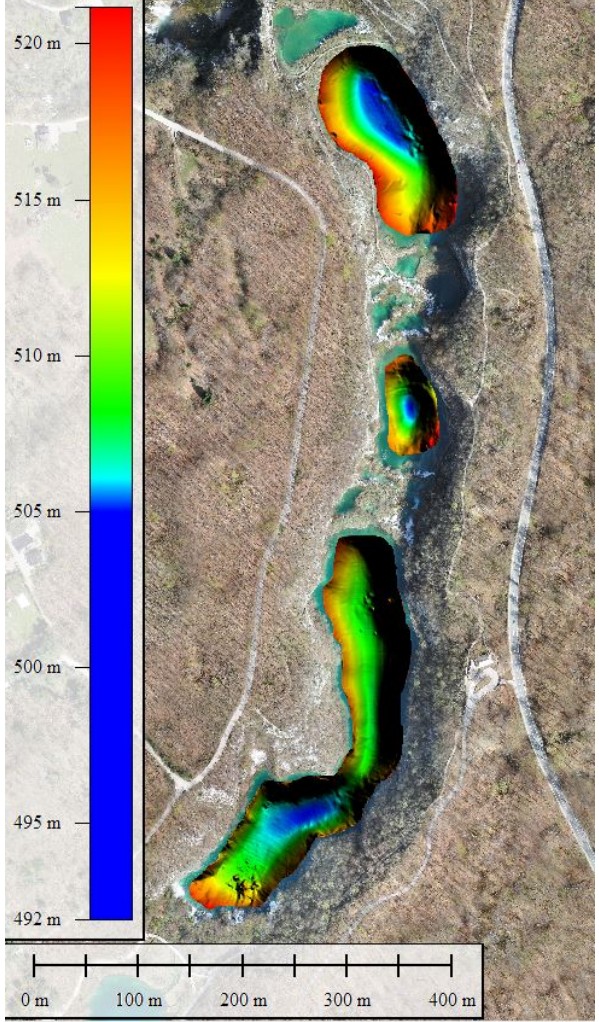

**Figure 13.** Lower Lakes' georeferenced bathymetric DEM layered over the digital orthophoto of the area. Top to bottom: Kaluđerovac, Gavanovac, and Milanovac lake.

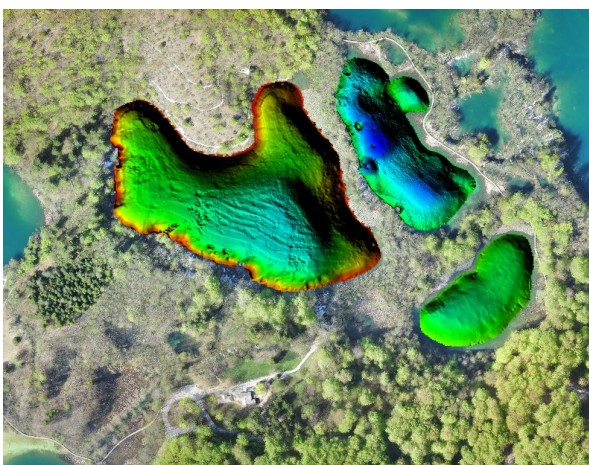

**Figure 14.** An example of the georeferenced digital elevation map of lakes Malo and Veliko jezero overlaid onto the orthophoto model created from the images taken by the UAV BEE-G3.

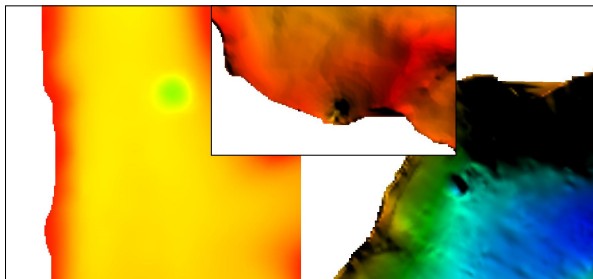

**Figure 15.** An example of depressions in lake beds due to the geologycal activity of tufa-made lake beds.

However, in order to be able to draw final conclusions, it is necessary to repeat the hydrographic measurements with the same and other instruments to make conclusions about the cause of the occurrence of anomalous depth values. Some of the depressions can be detected in bathymetric measurements. The analysis of the backscatter data, as shown in Figure 16 may reveal some anomalies that cannot be detected from bathymetric data because there is no expressed depression on the lake bottom. Furthermore, the difference in the intensity of the returned signal's intensity and the discernible circularity indicate a different sediment on the lake bottom. The use of a sub-bottom profiler or other underwater mapping solutions for the detailed mapping of structures under the lake bottom would also be of great benefit. This technology could further uncover the intricate connections between lakes with suspected underwater caverns and canals.

The raw data and the processed bahtymetry maps of the Plitvice National Park were then analysed and marked by the National Park staff. In the future, these bathymetric surveys will become a regular and periodic method for the geologists of the National Park to track changes in this geologically very dynamic environment.

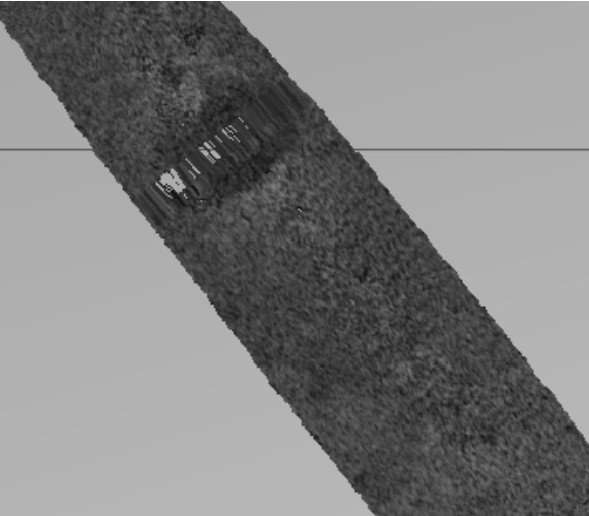

**Figure 16.** Details of one of the lakes' backscatter data mapped into 3D. An anomaly previously not detected from the bathymetric measurements is now clearly seen in backscatter data.

Table 1 summarises the data obtained through acoustic surveys of Plitvice Lakes, including the precisely measured altitude of the lakes' waterline, the maximum and average measured depth and the calculated area and volume. It is important to note that Petrik in [7] gathered in total 5400 limnological measurements of various types about all 16 lakes during a four-year period in the 1950s. Based on Table 1, the total number of data points processed (or so-called cells in the interpolated surfaces) in the QPS Qimera software is over 7.5 million for 13 lakes. This number is an order of magnitude lower than the number of raw sonar measurements. Collecting the measurements took 7 working days, and processing the sonar data took another 8 days, for a total of 15 days. This means that the state-of-the-art autonomous robotic systems, equipped with the latest visual and acoustic remote sensing technologies, enabled the researchers to collect over 1400 times more bathymetric measurements in a time frame more than 60 times shorter than that of [7].

**Table 1.** Summary of the data acquired by acoustic surveys of the Plitvice Lakes by the ASV PlaDyBath. Data for lakes Kozjak and Prošćansko are inferred from [16].

| Lake No. | Lake Name | Altitude [m] | Area [ha] | Max. depth [m] | Avg. depth [m] | No. Cells in Qimera |
|---|---|---|---|---|---|---|
| 1 | Prošćansko lake | 637.18 | 65.1 | 38.5 | 13.82 | N/A |
| 2 | Ciginovac | 626.6 | 6.5 | 13.29 | 9.5 | 1270.757 |
| 3 | Okrugljak | 614.1 | 3.4 | 10.87 | 6.59 | 826.249 |
| 4 | Batinovac | 609.8 | 1 | 5.48 | 3.35 | 134.273 |
| 5 | Veliko lake | 608.65 | 1.7 | 8.73 | 4.86 | 316.591 |
| 6 | Malo lake | 606.64 | 0.8 | 10.77 | 5.45 | 178.014 |
| 7 | Vir | N/A | N/A | N/A | N/A | N/A |
| 8 | Galovac | 585.2 | 11.8 | 24.66 | 11.75 | 3324.932 |
| 9 | Milino lake | N/A | N/A | N/A | N/A | N/A |
| 10 | Gradinsko lake | 553.75 | 6.2 | 9.79 | 4.81 | 824.172 |
| 11 | Burgeti | 553.68 | 0.5 | 11.2 | 6.67 | 130.748 |
| 12 | Kozjak | 535.48 | 77.3 | 47.5 | 17.61 | N/A |
| 13 | Milanovac | 524.26 | 3 | 19.21 | 12.89 | 405.181 |
| 14 | Gavanovac | 515.78 | 0.6 | 11.21 | 6.5 | 100.294 |
| 15 | Kaluđerovac | 506.44 | 1.4 | 14.49 | 8.72 | 214.94 |
| 16 | Novakovića brod | N/A | N/A | N/A | N/A | N/A |

## 6. Discussion

Plitvice Lakes National Park is protected under UNESCO's World Heritage List since 1979, so the requirements for any kind of data collection are remarkably high. The use of gasoline engines is strictly prohibited, so the only remaining option is electric propulsion. In addition, any kind of human intervention within National Park is forbidden, e.g., for the transport of remote sensing equipment and vessels, the removal of tree trunks fallen on the trails, vegetation in the vicinity and in the lakes. All these logistic constraints have led to the development of a small, two-person portable, electrically driven autonomous surface vehicle that can fulfil all the challenging tasks. The main advantages of this vehicle are its high portability, the ability to move omnidirectionally to follow highly rugged lake shores and the robustness to avoid potential hazards, such as branches and logs falling into the shallows of lakes, waterfalls, etc. Furthermore, the vessel is autonomous, so the bathymetric recording missions were carried out by the autopilot. The high-resolution orthophoto, which was generated based on the UAV's inspection of the entire area of Plitvice Lakes National Park, served as input for mission planning and to avoid potentially hazardous obstacles.

Images captured by the UAV can be processed using photogrammetry software to produce bathymetric maps of shallow lakes (less than 2 m) as well as near the edge of each lake where the ASV could not approach for safety reasons and multibeam sonar data was unavailable even after slanting sonar swath. In addition, the areas at the edge of the lakes are covered with dense thicket (marsh butterbur, sedges, willows). Upper Lakes are very inaccessible and the lake shores are surrounded by dense forests of fir and beech that overhang the lake shores. It is rather difficult to apply remote sensing methods in these areas, so these areas should be covered by a suitable direct measurement method. It would be necessary to cover these areas by remote sensing methods with a UAV at a lower altitude and with an oblique camera capture mode. The photogrammetry-based digital surface model could also be used for bathymetric mapping of the lake shallows. Based on the results from [43,44], an optimal method for correcting depth errors caused by water refraction could be chosen. It could then be merged with the acoustically recorded bathymetric maps of the deeper parts of the lakes to produce a complete DEM. This is an objective of authors' future work. Another possibility is to use a waterborne Ground Penetrating Radar (GPR) as in [45] or an airborne UAV-mounted GPR [46].

## 7. Conclusions

A great potential for the use of autonomous vehicles in remote sensing studies is presented in this article. Here an autonomous surface vehicle and an unmanned aerial vehicle were used for hydrology related autonomous remote sensing survey missions of Plitvice Lakes National Park in Croatia. The efficiency of using autonomous vehicles for such applications, the quality and quantity of data in time and the precision of georeferencing the data, especially in larger areas as in the authors' case study, has been shown to far exceed human performance. A total of 11 of 16 lakes in the National Park were surveyed using acoustic surveying methods with a multibeam sonar.

Interesting depressions at the bottom of the lakes were discovered in the bathymetric data, as well as in some previous research studies on Plitice lakes. It is assumed that these depressions in the lake bed are geomorphological features of the karst terrain that makes up the area. Reviewing and rerecording these anomalies, possibly even with a ROV or AUV for visual inspection, could have a major impact on understanding the intricacies of the Plitvice Lakes system. Their research will provide new insights into the system of lakes and its evolution. Previous studies have also reported such depressions but only a few of them. Now it has come to our attention that there are many more than expected still waiting to be explored.

The paper presents the scientific basis and methodology used in the modern autonomous robot geodetic measurements in the area of Plitvice Lakes National Park, its processing and the development of a digital three-dimensional geodetic models of lakes. These digital bathymetric maps will be the basis for further research purposes in GIS environment by experts in scientific disciplines such as biology, geology, hydrology, ecology, etc. Decisions on the way and form of protection of the underlying

phenomenon of tufa formation in Plitvice lakes would be made on the basis of these state-of-the-art maps. Based on the raw sonar ping data, snippets and backscatter or the entire unprocessed point cloud, a future objective of the authors is to develop methods for characterising the lake bed type. This, together with in-situ sampling, would lead to a definitive classification of the lake bed types.

Looking to the future, the authors' research objective is also to develop higher levels of vehicle autonomy, which would include autonomous exploration/inspection path (re)planning based on online processed sensor data and artificial intelligence to further optimise the performance of autonomous vehicles. If these investigations are carried out regularly, the geologists, biologists and ecologists at Plitvice Lakes National Park will be able to track changes of this very dynamic environment over time, based on high-resolution opto-acoustic models georeferenced with an accuracy of a few centimetres.

**Author Contributions:** Conceptualisation, N.K., A.V. and N.M.; Formal analysis, N.K. and B.K.; Methodology, N.K., B.K. and A.V.; Software, N.K., B.K. and Đ.N.; Supervision, A.V. and N.M.; Validation, N.K., B.K. and A.V.; Writing—original draft, N.K., B.K., A.V., Đ.N. and N.M. Writing—review and editing, N.K., B.K. and N.M. All authors have read and agreed to the published version of the manuscript.

**Funding:** This research is sponsored by Croatian Science Foundation Multi Year Project under G.A. No. IP-2016-06-2082 named CroMarX; the Foundation of the Croatian Academy of Science and Arts and the EU Regional Development funded project DATACROSS under G.A. No. KK.01.1.1.01.0009; the H2020-INFRAIA funded EUMarineRobots project under G.A. No. 731103; the European Regional Development Fund co-financed HEKTOR project KK.01.1.1.04; and Croatian Science Foundation co-financed GEOSEKVA project HRZZ IP-2016-06-1854.

**Acknowledgments:** The authors would also like to thank Milan Marković, the former mechanical engineer of LABUST, for the design and construction of the ASV PlaDyBath, as well as for all the help during the recordings of all the underwater archaeological sites in the scope of BLUEMED project; Nikica Kokir for all the technical and logistics help during data collection; Plitvice Lakes National Park staff for their help with the logistics and field work, as well as acoustic anomalies data interpretation; and Josip Barbača from Croatian Geological Survey for providing the geological map of Plitvice Lakes.

**Conflicts of Interest:** The authors declare no conflict of interest.

## Abbreviations

The following abbreviations are used in this manuscript:

| | |
|---|---|
| 2.5D | 2.5 dimensional |
| ASV | Autonomous Surface Vehicle |
| AUV | Autonomous Underwater Vehicle |
| CTD | Conductivity, Temperature and Depth sensor |
| DSLR | digital Single-Lens Reflex |
| DVL | Doppler Velocity Logger |
| GIS | Geographic Information System |
| GNSS | Global Navigation Satellite System |
| GNSS | Global Positioning System |
| GSM | Global System for Mobile Communications |
| IMC | Inter-Module Communication |
| IMU | Inertial Measurement Unit |
| LTE | Long-Term Evolution Communication Standard |
| QPS | Quality Positioning Services |
| ROS | Robot Operating System |
| ROV | Remotely Operated Vehicle |
| RTK | Real-Time Kinematic |
| SLAM | Simultaneous Localisation and Mapping |
| SSS | Side-Scan Sonar |
| UAV | Unmanned Aerial Vehicle |
| UCH | Underwater Cultural Heritage |
| USBL | ultra-short baseline |
| UUV | Unmanned Underwater Vehicle |

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
