# Peer review of "Autonomous Vehicles Mapping Plitvice Lakes National Park, Croatia"

_remotesensing, doi:10.3390/rs12223683_

Round 1
Reviewer 1 Report
The article presents the mapping approach of some lakes in Croatia. It is well structured however there is not a clear novelty on the presented work. Some remarks and suggestions are made directly onto the attached version of the article. However, some other are reported below:
In general, the projects that the Lab. is participating together with the available equipment, not related to this article is not of the reader's interest. It would be nice also to include a "contribution" section, describing the novelties of your work.
A "Discussion" section is missing.
An important note on UAVs for bathymetry is made in page 9. This note may help you in the discussion section also in order to cover the missing area between the UAS measurements and the shoreline.

Reviewer 2 Report
Excellent interesting application of integrating UAV and USV MBES for a challenging mapping task. Good description of all the systems and survey methodology and good presentation of the data. My comments are to enhance easier understanding of the paper.
Figure 2 is the key overall context for the reader but it is too complex and not readable even zooming on the screen. The country panel is not needed; it is covered in Fig 1. The geology map does not need all the sample lettering nor the detailed key. It is only the overall big units and the lakes that matter. The location of the lake sequence is what the reader wants. The numbering of the lakes is not visible. It would be useful to to clarify upper and lower lakes with "...the upper (southern)lakes..." and "..the lower (northern lakes ..."
Pg 3 Paragraphs "Today ..." and "Eutrophication..." need rewriting to clarify the balance of recent accelerating processes and long term natural processes. The text about trees is missing some words.
Pg 11 Para "If a lake..." Use figure 9 not "figure below"
Fig 10 is very hard to see the information, even zoomed. Can you remove the brown lake colour; it is not needed.
Section 4 pg 13 2nd para. Text about depth jumps and depth offset has some repetition or text missing.
Pg 14 Discussion about tectonic faults needs references or clarification of text. It seems to suddenly arise and not be adequately explained. Is it a detailed study of these Plitvice lakes or a general statement?
Fig 14, 15, 16. what is the black?
Round 2
Reviewer 1 Report
Authors addressed all my comments.